# Amplicon sequencing of pasteurized retail dairy enables genomic surveillance of H5N1 avian influenza virus in United States cattle

Andrew J. Lail[1], William C. Vuyk[1], Heather Machkovech[1], Nicholas R. Minor[2], Nura R. Hassan[1], Rhea Dalvie[1], Isla E. Emmen[1], Sydney Wolf[1], Annabelle Kalweit[1], Nancy Wilson[1], Christina M. Newman[1], Patrick Barros Tiburcio[1], Andrea Weiler[2], Thomas C. Friedrich[2,3], David H. O'Connor[1,2]*

1 School of Medicine and Public Health, University of Wisconsin, Madison, Wisconsin, United States of America, 2 Wisconsin National Primate Research Center, University of Wisconsin, Madison, Wisconsin, United States of America, 3 School of Veterinary Medicine, University of Wisconsin, Madison, Wisconsin, United States of America

* dhoconno@wisc.edu

## Abstract

Highly pathogenic avian influenza (HPAI) viruses with H5 hemagglutinin (HA) genes (clade 2.3.4.4b) are causing an ongoing panzootic in wild birds. Circulation of these viruses is associated with spillover infections in multiple species of mammals, including a large, unprecedented outbreak in American dairy cattle. Before widespread on-farm testing, there was an unmet need for genomic surveillance. Infected cattle can shed high amounts of HPAI H5N1 viruses in milk, allowing detection in pasteurized retail dairy samples. Over a 2-month sampling period in one Midwestern city, we obtained dairy products processed in 20 different states. Here we demonstrate that a tiled-amplicon sequencing approach produced over 90% genome coverage at greater than 20x depth from 5 of 13 viral RNA positive samples, with higher viral copies corresponding to better sequencing success. The sequences clustered phylogenetically within the rest of the cattle outbreak sequences reported. A combination of RT-qPCR testing and sequencing from retail dairy products can be a useful component of a One Health framework for responding to the avian influenza outbreak in cattle.

## Introduction

The One Health approach, as defined by the One Health High Level Expert Panel (OHHLEP), "recognizes the health of humans, domestic and wild animals, plants, and the wider environment (including ecosystems) are closely linked and interdependent" [1]. Recent epidemic and pandemic zoonoses have underscored the relevance and utility of One Health approaches to the study of emerging infectious disease [2,3]. The OHHLEP emphasizes the importance of communication, coordination, collaboration, and capacity building (the 4 Cs). Consistent with these guidelines, we developed

**Data availability statement:** Sequence data is available from the NCBI SRA database (accession numbers SRR31094760, SRR31094759, SRR31094758, SRR31094757, SRR31094756, SRR31094755, SRR31094754, SRR31094753, SRR31094752, SRR31094751). All other relevant data are within the paper and its Supporting Information files.

**Funding:** TCF and DHO received funding from The Institute for Clinical and Translational Research at the University of Wisconsin-Madison (https://ictr.wisc.edu/), grant 233 AAL9582. The funders did not play any role in the study design, data collection and analysis, decision to publish, or preparation of the manuscript.

**Competing interests:** I have read the journal's policy and the authors of this manuscript have the following competing interests: AJL has received travel funding from Oxford Nanopore Techologies to present this work at a conference. This does not alter our adherence to PLOS ONE policies on sharing data and materials.

an approach designed to build capacity for H5N1 genomic surveillance using retail milk as a pooled substrate, somewhat similar to wastewater, that is readily available for sampling, even when testing at individual farms or dairy processors is challenging.

In 2021, the HPAI clade 2.3.4.4b HA was introduced to North America as part of a multi-species panzootic affecting both birds and mammals, which has resulted in millions of wild bird and poultry deaths [4]. An ongoing outbreak of HPAI (H5N1) B3.13 genotype in dairy cattle was confirmed on March 25, 2024 by the National Veterinary Services Laboratories (NVSL) following reports of decreased milk yield from multiple states [5]. Phylogenetic analysis indicates this outbreak likely originated from a single introduction of an avian H5N1 virus into cattle, with subsequent spread facilitated by inter-farm movements of cattle, and potentially of people [6]. Exposure to cattle has resulted in multiple human cases of HPAI infection in dairy workers and dozens of feline HPAI infections [5,7,8]. Poultry workers have also been infected, underscoring the ability of this virus to cross species barriers [9].

HPAI infection in dairy cattle can occur within the mammary glands [5], resulting in milk titers up to $10^7$ $TCID_{50}$ per mL [10,11]. In commercial milk production, milk from cows on a farm is collected into a refrigerated temporary storage tank (bulk tank), then transported by a milk-hauling truck to a processing plant. Depending on the scale of the plant, milk from hundreds to thousands of animals may be combined. This milk then undergoes clarification, standardization, pasteurization (typically High Temperature Short Time pasteurization at 72°C for 15 seconds), homogenization, and packaging. Despite the dilution effect of pooling and potential degradation during pasteurization, HPAI viral RNA (vRNA) remains detectable in retail products. This is perhaps due to the known RNA stabilizing properties of milk [12,13].

The presence of HPAI vRNA in commercial pasteurized milk presents an opportunity for genomic surveillance. Several groups have used quantitative reverse transcriptase polymerase chain reaction (RT-qPCR) assays to test for HPAI H5N1 vRNA in commercial milk, observing geographic correlation to the known outbreak in cattle at the time [14,15]. Buying milk from grocery stores as a sampling method leverages the national distribution network of dairy products to provide a cost-effective, broad-scale surveillance strategy. Because pasteurization inactivates infectious virus, HPAI genomic surveillance using pasteurized dairy products can be performed in any BSL1 facility [16].

Although states like Colorado and California have implemented regular bulk-tank testing of herds regardless of clinical signs [17,18], most affected states did not have ongoing surveillance of farms in the absence of symptoms previous to the adoption of the USDA's National Milk Testing Strategy [19]. The lack of on-farm testing created a surveillance gap, hampering efforts to contain spread through early detection. Retail milk testing allows surveillance in regions without on-farm testing, but studies so far have focused more on testing for vRNA presence than on sequencing due to a lack of existing approaches for sequencing influenza from retail milk.

Here we aimed to determine whether we could detect and sequence vRNA in retail dairy products from states with known infections. We also sampled widely within Wisconsin (the second-largest dairy producing state in the US), where, as of

December 2024, there has not been a documented outbreak. Over a two-month period, we screened 66 commercial dairy products processed in 20 states using RT-qPCR. To sequence positive samples, we developed a tiled-amplicon approach to amplify fragmented influenza genome products. This tiled-amplicon approach was designed to work with our existing QIAseq Direct workflows for SARS-CoV-2 sequencing, enabling rapid response to a new pathogen of interest. Consistent with OHHLEP's 4 Cs, this strategy can provide a means to build capacity for HPAI genomic surveillance even when there is a lack of regional testing of individual farms or dairy processors [1]. Furthermore, because HPAI infectivity is inactivated by pasteurization, our approach does not require access to high-containment laboratories in which to process raw milk. This study demonstrates that retail dairy products are a readily accessible sample type for HPAI genomic surveillance in cattle, and provides an example of how the OHHLEP's One Health approach recommendations can be applied in the rapid response to an emerging zoonotic pathogen.

## Methods

### Sample collection

Sixty-six cartons of pasteurized retail dairy products (whole and low-fat milk, whipping cream, coffee creamer, yogurt, kefir, and buttermilk) were purchased from 11 different stores in the Madison, WI USA metro area from April 24 to June 17, 2024. Samples were purchased weekly or bi-weekly. Commercial dairy products are labeled with a plant code that includes the Federal Information Processing Standards code for the state where the dairy processing plant is located, combined with a unique number for that specific plant. This code can be used to trace the location of processing using the FDA's Interstate Milk Shippers list. The processing location does not necessarily correspond to the herd location, but it is the most accurate proxy for the herd location available at the point of sale. We aimed to collect whole fluid milk, but we also bought half and half, cream, kefir, and yogurt if whole milk was not available from a given plant code. We selected dairy products for RT-qPCR and sequencing based on processing plant code. Initially, samples were collected aiming for the broadest geographic sampling possible, with particular interest in states that had known active outbreaks. After successfully isolating and detecting vRNA, we aimed to resample from the same processing plants whose milk had detectable H5N1 HPAI vRNA in our initial testing. We also intensified our geographic focus on Wisconsin, testing as many processors as possible to enable early warning in this leading dairy state. We report results from a total of 29 different plant codes in this study.

We took 50 mL aliquots of product from the original container, recording the plant code, expiration date, and type of dairy product. Products that were too viscous to pipette were diluted 1:1 in nuclease-free water. All products were kept at 4°C throughout this process.

### RNA extraction

We have previously reported our protocol to extract viral RNA from milk on protocols.io [20].

We followed the MagMAX Wastewater Ultra Nucleic Acid Isolation Kit (Applied Biosystems) protocol on a Kingfisher Apex instrument (ThermoFisher Scientific) using 400 µL of neat or dilute dairy product as the input sample. Each sample was isolated in replicates of 6. Samples were eluted in 50 µL of elution buffer. We also isolated from 400 µL of water, which we treated as a sample in all downstream processing steps (Control A).

After isolation, one replicate of each extracted RNA sample was directly used for RT-qPCR, and the other 5 replicates were pooled for cleaning and concentrating. The pooled samples were incubated at 37°C for 30 minutes with 20 µL of 10X Turbo DNase buffer and 2 µL of TURBO™ DNase (Life Technologies). Then, samples containing vRNA were cleaned and concentrated using the RNA Clean and Concentrator-5 kit (Zymo Research) following the manufacturer's protocol, without the optional DNase I treatment. We also included a water control in this step (Control B). Since we used a large input volume, we increased the final elution volume from 15 to 30 uL. Eluted RNA was then used in the reverse transcriptase reaction for sequencing.

## Detection of HPAI using RT-qPCR

We used RT-qPCR to detect HPAI vRNA. Initially, we used a previously published assay targeting the influenza membrane protein (M) gene segment [21]. Any samples that tested positive (defined as a Ct less than 39) by the IAV M-gene assay were re-tested using the HPAI-specific assay described below.

For more specific HPAI detection, primer/probe sets were designed using Integrated DNA Technologies' PrimerQuest™ Tool (https://www.idtdna.com/pages/tools/primerquest) to target a consensus sequence of 6 randomly selected HA sequences from the HPAI outbreak in cattle (SRR28834851, SRR28834852, SRR28834853, SRR28834854, SRR28834855, SRR28834884) that we sourced from the Andersen Lab's avian-influenza github repository (https://github.com/andersen-lab/avian-influenza). The primer sequences were 5'-GGGAAGCTATGCGACCTAAAT-3' (forward) and 5'-CATTCCGGCACTCTGATGAA-3' (reverse). The probe sequence was 5'-ACATTGGGTTTCCGAGGAGCCATC-3'. The primers and probes were aligned *in silico* to ensure they would bind to all reported 2.3.4.4b HA currently circulating in birds and to ensure they were specific to 2.3.4.4b HA (S1 File).

Our RT-qPCR assays used TaqMan Fast Virus 1-Step Master Mix (Thermo Fisher Scientific) with primers at a final concentration of 600 nM and probe at a final concentration of 100 nM. The RT-qPCR was conducted using the LightCycler® 480 System (Roche Diagnostics) with cycling conditions of 50°C for 5 minutes, 95°C for 20 seconds, followed by 50 cycles of 95°C for 15 seconds and 60°C for one minute. Samples were tested in duplicate and considered positive when the average Ct was less than 39. For the HPAI-specific assay, absolute concentration within the reaction was determined by interpolation onto a standard curve with known concentrations of synthetic H5 double-stranded DNA manufactured and quantified by IDT. The standard curve ranged from $1.93 \times 10^6$ to 1.93 copies per reaction.

## Reverse transcription

We performed RT-PCR, amplification, and library prep as described in our publicly available protocol (https://dx.doi.org/10.17504/protocols.io.kqdg322kpv25/v1). We used reagents from the QIAseq DIRECT SARS-CoV-2 Kit A and QIAseq DIRECT SARS-CoV-2 Enhancer (Qiagen). For each sample, we assembled a reaction master mix on ice with RP Primer (1 μL), Multimodal RT Buffer (4 μL), RNase Inhibitor (1 μL), nuclease-free water (8 μL), and EZ Reverse Transcriptase (1 μL). We added 5 μL of cleaned RNA sample to this reaction. The thermal cycling conditions were 25°C for 10 minutes, 42°C for 10 minutes, 85°C for 5 minutes, followed by a hold at 4°C. At this point we also added a no-template control sample consisting of nuclease-free water (Control C). After reverse transcription, samples were taken directly to amplicon PCR.

## Amplicon generation

We generated a primer scheme for tiled amplicon sequencing using PrimalScheme [22]. As our input to PrimalScheme, we used 239 sequences cattle sequences (S1 Table) from the Anderson lab's Github repository (https://github.com/andersen-lab/avian-influenza), which in turn sources sequences from the USDA's collection of sequences related to the cattle outbreak (BioProject accession number PRJNA1102327 in the NCBI BioProject database, https://www.ncbi.nlm.nih.gov/bioproject/). Duplicate sequences within each segment were removed to avoid biasing. The amplicon size was 250 bp. Each segment was input separately. The algorithm did not identify primer sequences at the 5' and 3' termini of each segment, so we manually designed additional primers to capture the ends of each segment. At the time primer sets were generated there were no sequences available for the complete noncoding regions (NCRs) in any gene segments from H5N1 viruses causing the cattle outbreak, so we inferred likely NCR sequences based on publicly available sequences of clade 2.3.4.4b H5N1 viruses isolated from North American birds and mammals. We designed 5' and 3' terminal primers for each segment using approaches we have described previously [23], which have recently been adapted by Jaeger et al. [24]. The primer pools were solubilized in water to a stock concentration of 100 uM total, then diluted to a working concentration of 10 uM total.

## Amplicon sequencing PCR

We set up two polymerase chain reactions for each sample (one for each primer pool) using reagents from QIAseq DIRECT SARS-CoV-2 Kit A and QIAseq DIRECT SARS-CoV-2 Enhancer according to Table 1.

The cycling conditions were 98˚C for 2 minutes, then 4 cycles of 98˚C for 20 seconds and 63˚C for 5 minutes, followed by 29 cycles of 98˚C for 20 seconds and 63˚C for 3 minutes with a final hold at 4˚C. A Qubit fluorometer (ThermoFisher) was used to quantify the double-stranded DNA after the reaction.

## Library preparation and sequencing

We prepared our library according to the protocol for the Oxford Nanopore Technology Native Barcoding Kit V14 [25]. Nuclease-free water was included in the library preparation as a negative control for this step (Control D).

The library was sequenced on a minION mk1C sequencer using an R10 flow cell.

## Sequencing analysis methods

To perform Oxford Nanopore base-calling, primer-trimming, and variant-calling, we developed a bespoke Nextflow pipeline available at https://github.com/dholab/Lail-et-al-2024-analysis-pipeline [26]. The pipeline handles parallelizing tasks, running reproducible software containers via Docker, and providing a configurable command-line interface to the underlying tools it implements. These tools include the Oxford Nanopore Dorado base-caller v0.7.1, CutAdapt v4.8 for primer and adapter trimming, minimap2 v2.28 for read alignment, iVar v1.4.2 for variant-calling, and snpEff v5.2 for variant-effect annotation. Utilities in SeqKit2 v2.8.2, rasusa v2.0.0, vsearch v2.28.1, and samtools v1.5 are used throughout the pipeline, alongside other tools [27–34]. A full list of the pipeline's dependencies can be found in the GitHub repository's Python project configuration file, pyproject.toml, available at https://github.com/dholab/Lail-et-al-2024-analysis-pipeline/blob/main/pyproject.toml. In short, the core pipeline steps for each sample are: base-calling with Dorado, trimming reads by quality and selecting those reads that span complete amplicons, mapping to a reference genome, and consensus-sequence- and variant-calling.

Reads for this manuscript were mapped to A/Bovine/texas/24-029328-01/2024(H5N1), a sequence isolated from an infected cow's milk early in the outbreak and reported by Burroughs et al [8]. We added synthetic overhangs to our reference sequence to enable our primers to map to the reference. This is necessary since our end primers extend beyond the sequence available in the reference sequence. We used FLAN to annotate open reading frames within our custom reference to allow for mutations to be called at the amino acid level [35]. For our analysis parameters, we used the containerless profile with a minimum read length cutoff of 150 and a maximum read length cutoff of 500, along with a minimum read quality cutoff of 10. We allowed up to 2 primer mismatches between a given primer and the reference sequence. Variants and consensus sequences were called using a minimum required depth of coverage of 20 reads and a minimum variant frequency of 0.1.

Sequences were uploaded to Nextclade ([36], https://clades.nextstrain.org) for analysis against other publicly available sequences. HA gene segments were uploaded and analyzed against the H5Nx clade 2.3.4.4 tree (community/moncla-lab/

**Table 1. PCR set up for tiled amplicon strategy.**

| Reagent | Amount | Final Concentration |
| --- | --- | --- |
| cDNA from sample | 8 uL | |
| Primer pool, 10 µM | 4 uL | 1.6 uM |
| UPCR buffer, 5X | 5 uL | 1X |
| QN Taq Polymerase | 1 uL | |
| Nuclease-free water | 7 uL | |

iav-h5/ha/2.3.4.4). We used the dataset specific to the cattle outbreak (avian-flu/h5n1-cattle-outbreak/genome) to analyze the sequence of all 8 genome segments concatenated together. This dataset contains sequences from individual cattle as well as samples from other domestic and peri-domestic animals. Additionally, we used the Nextstrain H5N1 cattle outbreak tree to understand how common the mutations we detected were. We filtered by genotype at a specific amino acid and recorded how many sequences contained the mutation out of the total number of sequences in the tree. To expand this analysis to avian influenza sequences broadly, not just those related to the cattle outbreak, we used the same strategy to count mutations present in the segment-specific H5N1 trees from the past two years (e.g., for the polymerase-basic 2 (PB2) segment, avian-flu/h5n1/pb2/2y).

We used the Augur pipeline's traits function to predict the processing state of our samples [37]. This function is designed to provide an estimation of missing metadata based on metadata contained within the rest of a tree's nodes. To run the tool, we downloaded the H5N1 cattle outbreak Nextclade tree with our sequences included and used that as input to the traits function.

## Results

### RT-qPCR detects H5N1 vRNA in pasteurized dairy products

During a two-month surveillance period from April 24 to June 17, 2024, we tested 66 pasteurized commercial dairy products processed in 20 different states across the United States. At the end of the sampling period, the following states had outbreaks confirmed by NVSL: Texas, Kansas, Michigan, New Mexico, Ohio, Idaho, South Dakota, North Carolina, Colorado, Minnesota, Wyoming, and Iowa. Thirteen positive samples were identified in products processed from five states: Colorado, Indiana, Kentucky, Michigan, and Texas (S3 Table). It is important to note that the 20% proportion of positive samples is not unbiased; some producers' products were tested repeatedly upon identifying H5 genetic material. Three of the positives that we identified (Cartons 19, 27, and 41) were sequenced as part of our method optimization and performed poorly. We did not attempt to re-sequence them with the final sequencing protocol. Full RT-qPCR testing results are available in S3 Table.

Since our initial determination of positive and negative were based only on Ct values, we re-tested freshly isolated vRNA from all of our previously positive samples in a single RT-qPCR assay to quantify vRNA concentrations. In the interest of developing the assay quickly, we used double-stranded DNA as a standard rather than transcribed or viral RNA. The viral RNA copy numbers ranged from 1.92–4.50 $\log_{10}$ copies per mL of dairy product, with a median of 2.83 $\log_{10}$ copies per mL (Table 2).

### Tiled-amplicon sequencing covers all H5N1 gene segments

We developed a tiled amplicon approach to analyze H5N1 HPAI vRNA in pasteurized commercial dairy products. The method yielded an overall average sequencing depth of 8,176x across all samples, with individual samples varying from 325x to 29,256x. This correlated with the input vRNA amount, with less input vRNA yielding lower depth in sequencing. A Spearman's correlation using the scipy.stats tool gave a coefficient of 0.946, with a p-value of $3 \times 10^{-7}$ [38]. The average depth across all samples was lowest for the neuraminidase (NA) segment, with a depth of 199x. The average depth was highest for the polymerase-basic 1 (PB1) gene segment with a depth of 419x. Regions within the same gene segment may be sequenced to different depths because amplification efficiency may differ by amplicon (Fig 1). We used a 20x depth cutoff at a given position to determine the percent of the segment that was covered by our sequencing.

Coverage per segment ranged from 20% to 98% (Fig 2).

We considered any samples that had an overall coverage across the genome of greater than 90% in subsequent analyses. We chose this cutoff to allow for some amplicon failure while still retaining enough of the sequence to allow robust comparisons with other sequences. With this threshold we obtained passable sequences for every segment of carton 24. Carton 05, 46, 63, and 65 had multiple segments passing the quality threshold, and all of these samples exceeded 90%

**Table 2.  Geographic origin, qPCR results, and sequencing coverage for samples and controls sequenced in this study.**

| Sample Name | State | Purchase Date | Expiration Date | Log copies per mL dairy product | % Sequence coverage at 20X depth | Reads mapped |
|---|---|---|---|---|---|---|
| Carton 05 | CO | 4/24/2024 | 6/20/2024 | 2.89 | 90.0 | 331,732 |
| Carton 06 | CO | 4/24/2024 | 5/19/2024 | 2.57 | 84.5 | 236,964 |
| Carton 21 | TX | 5/2/2024 | 6/4/2024 | 3.09 | 89.2 | 272,222 |
| Carton 24 | MI | 5/2/2024 | 7/11/2024 | 4.50 | 95.7 | 1,950,789 |
| Carton 33 | CO | 5/20/2024 | 9/12/2024 | | 65.8 | 71,809 |
| Carton 36 | CO | 5/20/2024 | 7/8/2024 | 1.92 | 44.2 | 36,290 |
| Carton 46 | CO | 6/5/2024 | 8/16/2024 | 2.46 | 92.7 | 753,937 |
| Carton 48 | MI | 6/5/2024 | 8/22/2024 | 2.83 | 74.3 | 21,685 |
| Carton 63 | MI | 6/18/2024 | 8/29/2024 | 2.81 | 92.7 | 760,342 |
| Carton 65 | CO | 6/18/2024 | 8/23/2024 | 3.56 | 92.6 | 1,016,270 |
| Control A | | | | | 0 | 171 |
| Control B | | | | | 0 | 126 |
| Control C | | | | | 0 | 104 |
| Control D | | | | | 0 | 119 |

ªCarton 33 was initially tested by RT-qPCR during early optimization of our HPAI-specific assay, before we included a standard curve. Once we obtained a standard curve for the assay, we ran all the previously positive samples again to quantify viral copies per mL. In this second test, carton 33 was no longer positive. We still attempted to sequence it because of the initial assay result.

coverage genome-wide. Carton 21 was very close to the quality cutoff at 89%, along with carton 06 at 85%. Several individual segments in carton 21 were at greater than 90% coverage, but only one segment from carton 06 (NA) reached this threshold. No individual segments from cartons 33, 36, or 48 had coverage at over 90%.

## Sequences cluster within reported outbreak sequences

Initially, we confirmed that our H5 consensus sequences were clade 2.3.4.4b HA by clustering with the H5Nx clade 2.3.4.4 dataset. All of our HA sequences were contained within the clade 2.3.4.4b lineage (S1 Fig).

We also wanted to confirm that our consensus sequences were contained within the diversity of other sequences from the HPAI outbreak in cattle. All of our samples clustered within the known diversity of the current North American H5N1 outbreak (Fig 3). We anticipated that our sequences would cluster closely with sequences from their processing plant state. For example, we expected viruses in milk processed in Michigan to cluster with outbreak sequences from Michigan. We masked our state-level metadata and used Augur's "traits" function to infer the original processing state [37]. It provided the correct state for three out of five samples.

## Few major amino acid substitutions were detected

We next identified major (consensus-defining) and minor (sub-consensus) mutations present in each sequence we generated, relative to the A/Bovine/texas/24-029328-01/2024(H5N1) reference. It should be noted that this reference sequence is from the cattle outbreak, so concerning mutations present throughout the cattle outbreak (such as HA:172A, NA:55T, NA:67I, PB2:495I, and PB2:631L) were not highlighted in this analysis [6]. We also compared the mutations we found to previously published lists of concerning mutations in influenza A [6,39,40]. We detected a total of 32 major mutations across all our samples, of which 8 were non-synonymous (Table 3). We did not detect non-synonymous mutations in HA in any of these sequences. A major mutation we observed was PB2:A255V, which was present in all five samples. We observed this mutation relative to our reference, which is one of a few examples in the cattle outbreak that contains an alanine at PB2:255. We identified PB1:V171M in two of our sequences, a relatively rare mutation through the rest of the

**Fig 1. Genomic coverage of samples at 20X depth.** Coverage at each base is plotted per-segment, with each color line representing a different carton. The vertical axis is given in log base 10, with a dashed line at $\log_{10}(20)$ to indicate our depth cutoff. The sharp drop-offs in coverage depth that dip below the dashed line correspond to amplicons that did not amplify or sequence well. Gene segments are abbreviated as follows: polymerase-basic 2 (PB2), polymerase-basic 1 (PB1), polymerase acidic (PA), hemagglutinin (HA), nucleoprotein (NP), neuraminidase (NA), membrane protein (MP), and nonstructural (NS).

cattle outbreak. PB1:V171M is much more common throughout avian influenza sequences worldwide (72%). In PA, we found PA:Y24H, PA:A36T, and PA:I596V, all of which are present in less than 0.5% of the rest of the cattle sequences. Of these, PA:A36T has been noted as a potentially virulence-enhancing mutation in mice [41]. We identified NA:S71N in two of our sequences. This mutation is present in over 30% of cattle outbreak sequences and is associated with modulating host-range [6].

We used a threshold of 10% frequency to identify any sub-consensus mutations present in our sequences. Out of a total of 29 sub-consensus mutations present in our passing sequences, there was only one sub-consensus mutation of note. We observed HA:P324L (H3 numbering) in carton 24 at a 36% frequency, which is a mutation implicated in increased virulence [6].

## Discussion

The HPAI H5N1 2.3.4.4b virus currently spreading in dairy cows is a prime example of pathogen exchange between wild animals, domestic animals, and humans, as the One Health concept reminds us [1]. The outbreak is concerning due to its potential for mammalian adaptation and spillover while circulating in a domestic mammal with close proximity to humans. Furthermore, there is still a need for expanded genomic surveillance. Although only 17 states have reported dairy herds infected with HPAI H5N1 viruses (as of March 2025), those states are distributed throughout the US. In the past, testing

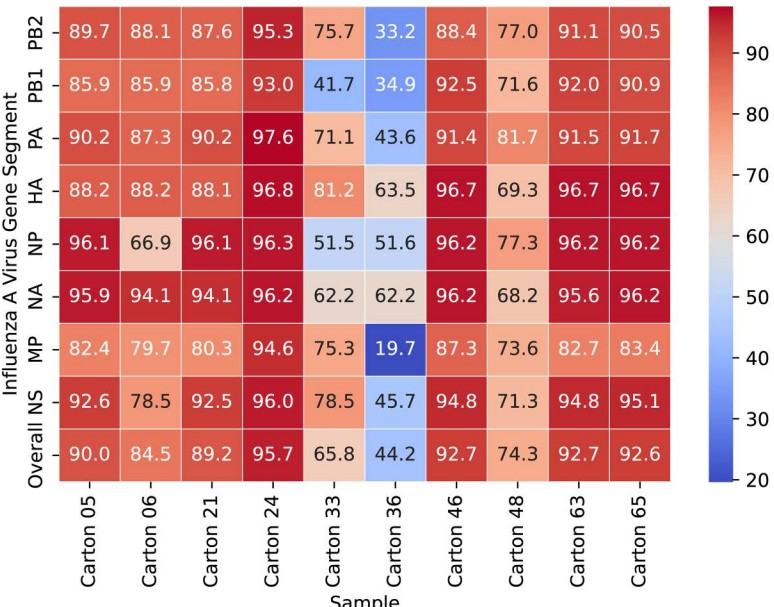

**Fig 2. Percent coverage at 20x depth by segment for each sample.** The red colors correspond to higher coverage, while the blue colors correspond to lower coverage. All numbers are expressed as percentages.

has been performed at different frequencies in different states. With the start of the National Milk Testing Strategy, we hope that sporadic, state by state testing will soon transition to regular, comprehensive surveillance by the USDA [19].

Retail dairy sampling is an orthogonal approach to current bulk tank testing efforts. Previous studies established that HPAI H5N1 vRNA can be detected in retail dairy products using RT-qPCR. Here we show that such retail dairy products can also be used to generate near-complete HPAI H5N1 viral genome sequences suitable for phylogenetic and evolutionary analyses. We did not uncover any new concerning mutations in our limited sampling, but if this approach were expanded it may enable increased genomic surveillance in an environment of limited individual animal testing. In line with the OHHLEP's 4 Cs (communication, coordination, collaboration, and capacity building), we have shared all of our protocols and results publicly throughout this study such that researchers at any properly equipped BSL-1 facility could obtain retail dairy samples from a local grocery store and contribute to HPAI genomic surveillance.

## Limitations of this approach

Only a small fraction of nationwide dairy production is allocated to fluid milk. According to the USDA Economic Research Service, "most of the milk supply is used to produce manufactured dairy products" like cheese and butter [42]. For example, in Wisconsin 90% of fluid milk is used for cheese production [43]. This reduces the effectiveness of retail milk sampling for surveillance purposes because some farms and dairy processors produce little or no product that could be sampled as retail fluid milk. We have attempted to detect and sequence H5N1 vRNA in other dairy products, but all of those have potential pitfalls. Many manufactured dairy products contain multiple dairy ingredients, like cream, milk powder, or whey. These ingredients may be sourced from multiple geographically distant processing plants, obfuscating the original geographic origin of the dairy material containing HPAI H5N1 RNA. We operate under the assumption that retail milk is likely processed at a plant that is close to its origin farm, since it is not economically viable to haul milk over long distances [44]. While a sample that tests positive from an area is an indicator of H5N1 infection, the reverse is not true.

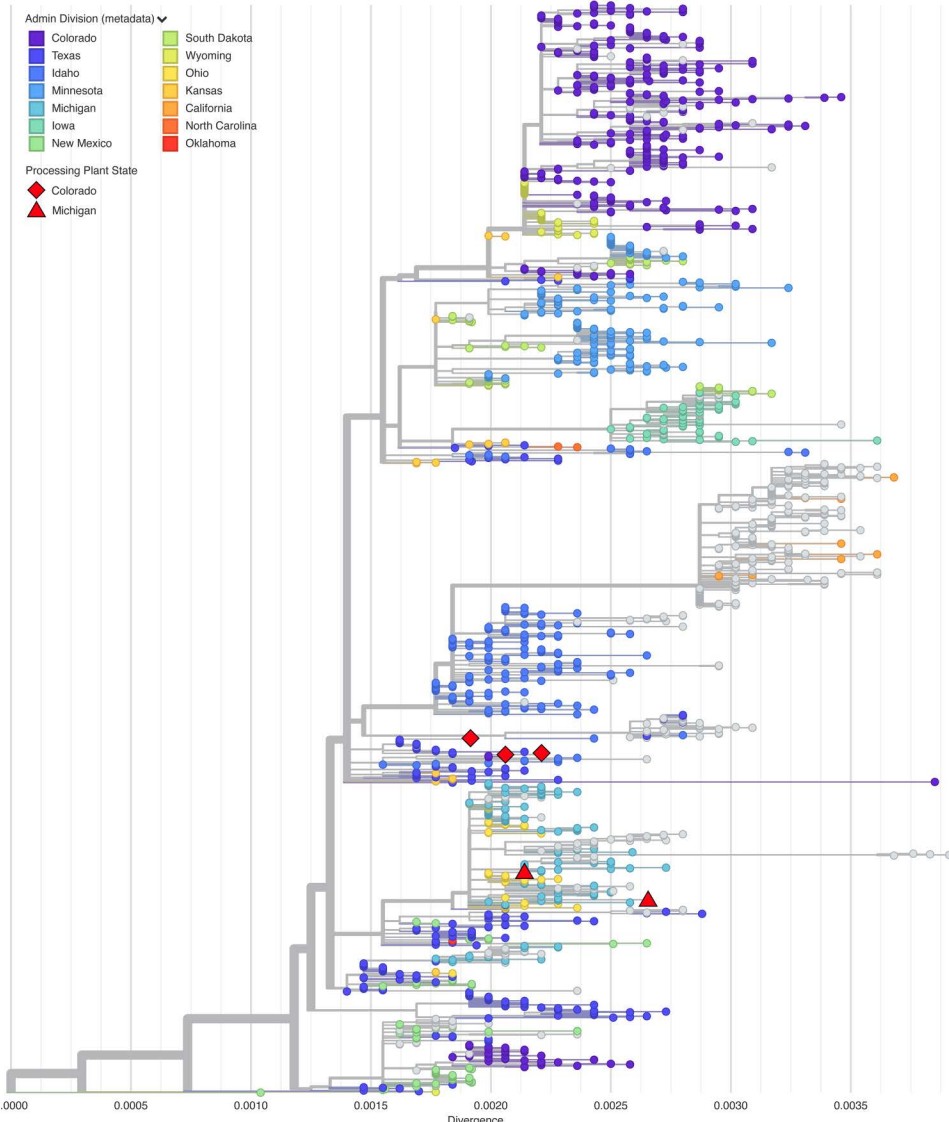

**Fig 3. Phylogenetic clustering in Nextclade.** The consensus sequences from each dairy product (represented by red diamonds and triangles) are superimposed on the background tree sourced from Nextclade (https://nextstrain.org/avian-flu/h5n1-cattle-outbreak/genome). Samples are colored by their admin division as reported in the sequence metadata, with missing metadata represented by gray color. The x-axis shows divergence, a measure of genetic distance between sequences (approximately number of nucleotide substitutions per nucleotide position).

There are many reasons why H5N1 infections may not be picked up by this sampling strategy, so a negative test from a given area is not a declaration that influenza is absent in that area.

Additionally, consensus sequences obtained from retail dairy may represent viruses from a single cow or, more likely, multiple animals. This limits the utility of phylogenetic analyses, which assume that each sequence represents a single infected individual. Various approaches have been used to deconvolute inputs from multiple infections in other pooled samples, but given the short length of our reads and the relatively low genetic diversity within the current HPAI H5N1 cattle outbreak, these are not helpful for deconvoluting our dataset [45–47]. The difficulty of deconvoluting the tiled-amplicon reads is a major downside of our approach compared to whole-segment sequencing.

**Table 3. Amino acid variants at the consensus level in each sample.**

| Gene | AA Position | Reference | Carton 5 | Carton 24 | Carton 46 | Carton 63 | Carton 65 | % of Nextstrain H5N1 cattle outbreak sequences with mutation |
|------|-------------|-----------|----------|-----------|-----------|-----------|-----------|------------------------------------------------------------|
| PB2 | 255 | A | **V** | **V** | **V** | **V** | **V** | 99.38% |
| PB1 | 171 | V | V | V | **M** | V | **M** | 0.19% |
| PA | 24 | Y | Y | Y | **H** | Y | **H** | 0.43% |
| PA | 36 | A | A | A | A | **T** | A | 0.12% |
| PA | 596 | I | I | I | I | **V** | I | 0.00% |
| NS | 79 | M | M | M | M | **I** | M | 0.25% |
| MP | 28 | I | **T** | I | I | I | I | 5.49% |
| NA | 71 | S | S | **N** | S | **N** | S | 30.29% |

These variants are relative to A/Bovine/texas/24-029328-01/2024(H5N1).

In our phylogenetic analysis, HPAI sequences from retail dairy samples were not always part of the same lineage of HPAI sequences as infected cattle from the same state. There are several potential explanations for this. Due to the fairly recent common ancestor of the outbreak, the existing variation within the HPAI outbreak in cattle is constrained, limiting our ability to confidently assign lineages. In addition, while some portion of the HPAI cattle sequences do cluster by state, many sequences do not. Moreover, there may be movement of virus among states, increasing within state virus diversity (reducing between state diversity) and reducing our ability to accurately identify state of origin for a given sample. Movement of virus between states was more likely to occur before interstate testing requirements were introduced [48]. Because our samples were all purchased within 2 months of those requirements being introduced, it is possible that interstate transmission makes our samples more difficult to place in this analysis.

While the tiled amplicon approach allows us to sequence more degraded samples, it comes with limitations. The approach is designed to work on B3.13 clade sequences, and it is unlikely to work as well in the presence of recombination.

## Open science for emerging threats

Our team's ability to quickly adapt to sequencing HPAI from the emerging cattle outbreak was due in part to our ability to pivot existing tools, protocols, and collaborations established for SARS-CoV-2 genomic monitoring projects. We repurposed the reagent kits we had on hand for SARS-CoV-2 sample processing and sequencing (QIAseq Direct). This allowed us to process new samples almost immediately with protocols that were already established. Additionally, since retail dairy sampling can be done in BSL-1 facilities, this work could be democratized to laboratories that do not have containment facilities required to handle pathogenic HPAI H5N1.

Public data sharing networks strengthened by the COVID-19 pandemic facilitated our use of initial outbreak sequences posted by colleagues. Access to these sequences was necessary for the rapid generation of a whole-genome tiled-amplicon primer scheme with PrimalScheme, which allows for automatic generation of a tiled-amplicon primer scheme of arbitrary length based on a reference fasta sequence [22]. Shared data analysis platforms like Nextstrain, and the data submitters that power them (S5 Table) were also key to understanding the sequencing results.

Transitioning from SARS-CoV-2 sequence surveillance to HPAI sequence surveillance speaks broadly to the importance of maintaining inter-pandemic expertise, the use and development of flexible multi-pathogen lab workflows, and networks of open data and protocol sharing, as has been encouraged by the OHHLEP and the CDC SPHERES Consortium [1,49].

Given the diverse geographic origins of milk available at our local grocery stores, it is possible that additional sampling hubs could sample the majority of retail fluid milk processors nationwide. There are several states with high milk production that our group has been unable to sample rigorously, like California, Texas, Idaho, and New York. This concept is already being demonstrated in Canada, where a network has been formed to sample retail milk longitudinally to detect HPAI in dairy cattle [50]. In fact, sampling retail milk to detect cattle outbreaks could be implemented in any area where the production region of milk is available at the point of sale, not just in Canada and the US.

Finally, it is worth considering what other types of environmental sampling might be valuable for monitoring future zoonotic risks posed by livestock. Swine are often considered a mixing vessel for recombination between bird, human, and swine flu strains [51]. This is reflected in the strict biosecurity protocols in use on pig farms. Just like dairy farms, these operations have retail outputs, like meat, which may prove helpful to monitoring these viruses. Performing viral genomic surveillance on asymptomatic animals at the population level by sampling outputs from these locations may generate data to help inform risk assessments. However, it is unlikely that the same approach used for milk samples in this study can simply be reconfigured to swine or another system. There is a need for quick turnaround and traceable supply chains to produce geographically informed, actionable information. Milk is pooled between many animals, often sold within a month of production, and contains a marker of where it was produced. It is possible that extending population level monitoring to other agricultural settings would require alternative approaches like wastewater or air monitoring.

Would consistent environmental sampling of wastewater or air at pig farms be helpful for detecting new variants of avian influenza viruses? What other grocery store or restaurant products could be used for pathogen monitoring? The unexpected detection of HPAI H5N1 in retail milk and the utility of this data for viral sequencing highlights how unorthodox approaches to genomic epidemiology can complement conventional "shoe-leather" epidemiology based on testing individual animals.

## Supporting information

**S1 Fig. Nextclade clustering of HA segment against the H5Nx 2.3.4.4 clade tree.** The sequences from this study are in the red box on the lower right side of the chart, branching out from the 2.3.4.4b subclade.
(TIF)

**S2 Fig. Colorado sequences (Carton 46 and 65) relative to nearest neighbors among published sequences.**
(TIF)

**S3 Fig. Carton 05 relative to nearest neighbors among published sequences.** Carton 05 is from Colorado. It clusters locally with feline and bovine sequences from Texas.
(TIF)

**S4 Fig. Michigan sequences from this study relative to nearest neighbors among published sequences.** Cartons 24 and 48 (red circles) cluster within a clade of mostly Michigan and Ohio sequences.
(TIF)

**S1 Table. Sequences from Andersen Lab github used for generating tiled-amplicon scheme.**
(XLSX)

**S2 Table. Primer pools used in tiled-amplicon sequencing.**
(XLSX)

**S3 Table. Full results from RT-qPCR testing.**
(XLSX)

**S4 Table. SRA Accession numbers for sequences produced in this study.**
(XLSX)

**S5 Table. Acknowledgement to sequence contributors in Nextclade.**
(XLSX)

**S1 File. Design of H5 HA RT-qPCR primers.**
(PDF)

## Author contributions

**Conceptualization:** Thomas C. Friedrich, David H. O'Connor.

**Data curation:** Andrew J. Lail, William C. Vuyk, Nicholas R. Minor, Nura R. Hassan, Rhea Dalvie, Isla E. Emmen, Sydney Wolf, Nancy Wilson.

**Formal analysis:** Andrew J. Lail, Heather Machkovech, Nicholas R. Minor.

**Funding acquisition:** Thomas C. Friedrich, David H. O'Connor.

**Investigation:** Andrew J. Lail, William C. Vuyk, Heather Machkovech, Nura R. Hassan, Rhea Dalvie, Isla E. Emmen, Sydney Wolf, Patrick Barros Tiburcio.

**Methodology:** Christina M. Newman, Andrea Weiler.

**Project administration:** Nancy Wilson.

**Resources:** Thomas C. Friedrich, David H. O'Connor.

**Software:** Andrew J. Lail, Nicholas R. Minor, Annabelle Kalweit, David H. O'Connor.

**Supervision:** Thomas C. Friedrich, David H. O'Connor.

**Visualization:** Andrew J. Lail, Nicholas R. Minor, Annabelle Kalweit, Nancy Wilson.

**Writing – original draft:** Andrew J. Lail, William C. Vuyk, Heather Machkovech.

**Writing – review & editing:** Andrew J. Lail, William C. Vuyk, Heather Machkovech, Nicholas R. Minor, Nura R. Hassan, Rhea Dalvie, Isla E. Emmen, Sydney Wolf, Annabelle Kalweit, Nancy Wilson, Christina M. Newman, Patrick Barros Tiburcio, Andrea Weiler, Thomas C. Friedrich, David H. O'Connor.

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
