## [Decision Letter · Decision Letter 0]

29 Jan 2025

Dear Dr. Lail,

Thank you for submitting your manuscript to PLOS ONE. After careful consideration, we feel that it has merit but does not fully meet PLOS ONE’s publication criteria as it currently stands. Therefore, we invite you to submit a revised version of the manuscript that addresses the points raised during the review process.

We look forward to receiving your revised manuscript.

Kind regards,

Ayi Vandi Kwaghe, D.V.M., M.V.Sc., P.G.D.E. Ph.D., MPH, FETP

Academic Editor

PLOS ONE

Journal Requirements:

“I have read the journal's policy and the authors of this manuscript have the following competing interests: AJL has received travel funding from Oxford Nanopore Techologies to present this work at a conference.”

Reviewers' comments:

Reviewer's Responses to Questions

**Comments to the Author**

1. Is the manuscript technically sound, and do the data support the conclusions?

Reviewer #1: Partly

Reviewer #2: Yes

Reviewer #3: Yes

Reviewer #4: Yes

Reviewer #5: Yes

Reviewer #6: Yes

Reviewer #7: Partly

2. Has the statistical analysis been performed appropriately and rigorously?

Reviewer #1: N/A

Reviewer #2: N/A

Reviewer #3: Yes

Reviewer #4: Yes

Reviewer #5: Yes

Reviewer #6: No

Reviewer #7: No

3. Have the authors made all data underlying the findings in their manuscript fully available?

Reviewer #1: Yes

Reviewer #2: Yes

Reviewer #3: Yes

Reviewer #4: Yes

Reviewer #5: Yes

Reviewer #6: Yes

Reviewer #7: Yes

4. Is the manuscript presented in an intelligible fashion and written in standard English?

Reviewer #1: Yes

Reviewer #2: Yes

Reviewer #3: Yes

Reviewer #4: Yes

Reviewer #5: Yes

Reviewer #6: Yes

Reviewer #7: Yes

Reviewer #1: This manuscript outlines an interesting protocol for testing milk products for the presence of H5N1 in cattle. It is not clear from the data presented what the sensitivity and specificity of the technique might be. This information is needed to assess the value of the protocol from a diagnostic perspective, Please clarify the statement '....Human infections have also been observed in poultry workers after presumed spillover of HPAI into poultry from cattle, underscoring the zoonotic potential of this outbreak (9).- currently it does not make sense and is not supported by the reference https://www.cdc.gov/mmwr/volumes/73/wr/mm7334a1.htm. As presented, this study does not fully align with a 'One Health approach' which is usually considered to be a transdisciplinary integrated approach to understanding a complex problem vs a focused study on a technique applied to detecting a potential pathogen which is zoonotic. There is mention in the conclusions about further sampling of pig farms to detect AIVs but no real consideration of how such data could be used in epidemiological studies and risk assessments. Can this be further discussed ?

Reviewer #2: The manuscript entitled “Pasteurized retail dairy enables genomic surveillance of H5N1 avian influenza virus in United States cattle” is a comprehensive short study of the detection and molecular characterization of HPAIV H5N1 in retail milk in the USA. The study is descriptive and well written, it brings in a new protocol for NGS of retail milk and confirms a surveillance perspective: looking at bulk milk rather than at individual cattle heads would allow for a faster, cheaper and more global detection and characterization of dairy cattle HPAIV H5N1. I just have the following questions/suggestions:

- There seem to be discrepancies between numbers in the text and tables. While it is clear that the authors have tested 66 cartons in 2 months, positives are described as coming from 5 states in the abstract/text but seem to come from just 3 in Table 2. While 18 cartons are positive according to table S3, 10 are positive in Table 2 and 13 described as positive in the text. Please clarify. The Ct value indicated for carton 28 (68.48) also seems off.

- How did you set your thresholds? Both for the 20x sequence coverage chosen and for the 10% frequency for the variants search.

- Mammalian adaptation markers are known. It is a pity that the authors did not screen their sequence data for those (irrespective of the variants frequency actually, it would be quite informative in a One Health perspective as stressed in the Introduction and Discussion sections).

- Figure 2: the tree is not very informative. What do divergence percentages refer to? What is the scale? The Figures legends are very short and would deserve more info.

Reviewer #3: The manuscript presents a novel and impactful approach to H5N1 surveillance using retail dairy products, aligning with the One Health framework and providing valuable insights into the spread of the virus in U.S. dairy cattle. However, the study has some limitations, including sample representativeness, phylogenetic challenges, and technical difficulties in sequencing low-abundance viral RNA. Addressing these limitations in future work could enhance the utility of retail dairy surveillance for monitoring zoonotic pathogens. Overall, the study is a significant contribution to the field of genomic epidemiology and has important implications for public health and food security.

Authors may improve the discussion section by discussing the global implications of the findings and how the approach could be adapted for use in other regions or for monitoring other zoonotic pathogens. Also, they have to clearly articulate the limitations of using retail dairy for surveillance, such as sample representativeness, phylogenetic challenges, and potential for false negatives.

Reviewer #4: I think this study fills an important gap In One Health approach to an outbreak response considering the challanges to cooperate with farm facilities which are the sourse of an ongoing infections locally and nationally. The study shows efficient traceability of a viral disease in readlily available milk samples using their resources in the lab facility. They not only detected viral RNA in milk samples using RT-QPCR but also developed a method to amplify and sequence the genome of the virus to discover mutations and understand dispersal of the virus. Although the method is not perfect, the authors discuss potential limitations of their study. Their methodology could be applied to sample dairy products rather than just milk in future studies.

Reviewer #5: Thank you for wonderful research and data which you present on this paper. I understand how difficult it is to collect the data and summarize it as it might be difficult to track whether the sequence is from vRNA of single dairy milk or from vRNA of some dairy milks.

Therefore, I have some questions and recommendations regarding this paper:

1. Paragraph 6, you don’t need to mentioned “the home state of the researchers on this study”, instead you might need to give more other interesting point about Wisconsin besides “the second- largest diary producing state in the US”.

2. It is better to put a reference about the existing workflow from your work in SARS-CoV-2 sequencing, as you mentioned it in introduction.

3. It is better to give reference to “OHHLEP’s 4 Cs” strategy which you also mentioned it in introduction.

4. You mentioned in “sample collections” about the dairy products which you took as sample, can you explain about the percentage of each sample taken? How many percent of whole milk did you take and how many percent low-fat wilk did you take? Etc.

5. It is unclear about how you finally only mentioned about samples from 10 sample cartons from 66 samples which you previously mentioned.

6. Why did the observation only do in cartons 21,24, 46,63 and 65? Because we could see that segments’ coverage on cartons 5 and 6 are also over than 90%.

7. The discussion part is more like simple introduction rather than a discussion of result which you have.

Reviewer #6: Summary:

This study describes detection and characterization of H5N1 viral RNA from commercial pasteurized dairy products. This is very timely as the H5N1 outbreak in wild and agricultural animals and resulting spillover infections in humans are of increasing and high concern. This study finds that H5N1 can be detected and sequenced from commercial milk, showing a readily accessible data source for genomic surveillance efforts. While results described are somewhat limited and figures could use improvement, especially figure 2, the interest and timeliness of the study are high. Further, the efforts towards methodological and data openness and utilizing tools developed in other outbreaks (SARS-CoV-2) demonstrates an important framework for the H5N1 and future outbreak responses.

Major Comments:

The abstract could use a more comprehensive summary of results – e.g. qPCR results, more description of genome coverage (significant correlation with qPCR result?) and/or phylogenetic clustering of resulting sequences.

A Spearman correlation should be performed to support the following claim “The method yielded an overall average sequencing depth of 8,176x across all samples, with individual samples varying from 325x to 29,256x. This correlated with the input vRNA amount, with less input vRNA yielding lower depth in sequencing.”

The quality of the Fig. 2 phylogeny needs to be improved for readability/interpretation. If a tree at higher resolution with less node crowding cannot be made in Auspice, recommend downloading the tree and metadata files from Auspice and re-creating the tree. Several tree tips in Fig. 2 are white/gray – do these not have location metadata? Besides trait inference, did sequenced samples cluster with other specimens from the state? It appears possible for Michigan samples but not Colorado but given the low quality/crowding in the tree is difficult to see. A description of clades that sequenced samples fall into may help.

Further description of sub-consensus mutations could be informative as the samples are pooled from many animals and detection of low-frequency mutations could indicate whether H5N1 was derived from one or many individuals.

Minor Comments:

The caveat that the H5 qPCR standard curve was dsDNA, which is not ideal for RNA virus absolute quantification should be mentioned.

“resulting in viral titers in milk of at least 107 TCID50 per mL (10).” should be “milk titers up to 107 TCID50 per mL”. Could also cite https://www.nature.com/articles/s41586-024-07849-4

Cited preprint 13 is now published and should be updated, other preprints should be checked for publication status

The last paragraph of the introduction contains text more appropriate in the results and discussion sections, consider revising.

Was the Ct cutoff of 39 based on limit of detection or other assay testing?

Table 3 could be converted to a heatmap to improve readability

Mutation PB2:M631L is common in H5N1 sequences from dairy cattle. Was this mutation not prevalent during the screening time described here? Why this mutation was not observed should be explained.

Putative reasons for repeated sampling of rare mutants could be discussed (PB1:171, PA:24). Are the sequences used in “% of nextstrain H5N1 cattle outbreak sequences with mutation” largely from a different source, e.g. could these be mutations associated with milk? A brief description of the referenced nextstrain sequences would be informative : # of sequences, collection dates and locations.

In S3 table, dholab_carton_0028 result is 68.48, is this an error (50 cycles reported in methods)?

Reviewer #7: Summary

The manuscript by Lail et al entitled, “Pasteurized retail dairy enables genomic surveillance of H5N1 avian influenza virus in United States cattle” reports the detection and whole genome sequencing of H5N1 high pathogenicity avian influenza viruses in pasteurized milk purchased in Madison, WI. The authors reported 18/67 cartons that were positive by qRT-PCR and 10 nearly complete genomes that were submitted to a public database. A tiled amplicon approach was used by the investigators to achieve sequencing of genomes from pasteurized milk.

General comments

• Whole genome sequencing for influenza viruses have typically been done using a universal primer approach that takes advantage of the conserved non-coding regions of influenza viruses at the end of the genomes. These contain packaging signals. A main advantage of the universal primers is that it can work for all known influenza A viruses. However, this approach also amplifies defective genomes as it only requires annealing to the ends of the genome. As a result, longer segments tend to have poor coverage in the middle.

• Potentially, the tiled approach the authors report has an advantage of having even coverage. Unfortunately, there appears to be sections where coverage drops precipitously (Figure 1). The drops in coverage appear to be 250 bp and is thus, most likely a failure of a primer set to amplify its target.

• It would be highly beneficial to compare the tiled method to universal primer set approach. A comparison would enable future users to choose the most appropriate genome amplification approach.

Specific comments

• I suggest a title change to focus on the sequencing method since the sampling was relatively limited and based on convenience (non-systematic). The major strength of the paper is the use of a novel approach to sequencing influenza viruses. However, the title suggest that systematic surveillance was conducted throughout United States whereas retail dairy products were only purchased in Wisconsin.

• Can you comment on the timings of collection/purchases? Was milk purchased weekly, daily, etc? Was each carton a distinct brand (No need to mention brand/trade names. Please anonymize)?

• Please report proportion of samples positive by qRT-PCR since this will be of interest to readers.

• From the genomes sequenced, how does it compare to other isolates from the same state?

• “Because pasteurization neutralizes infectious virus…”

o It would be more appropriate to use the term “inactivates” instead of “neutralizes” since neutralization of virus commonly refers to the preventing infectious virus to bind to host cells, whereas inactivation is a general term that refers to eliminating the ability of viruses to infect a host.

• What is the rationale behind a 20X cutoff for read depth?

• Data in Table 3 can be made more digestible with a graph

• Throughout the paper, please report log copies/mL instead of copies/mL. It is typical for viral RNA titers to be reported in the logarithmic scale because differences in orders of magnitude are more biologically relevant.

• What is the rationale for the A/bovine/Texas/24-029328-01/2024? Where is this strain located in the phylogenetic tree? While it is temporally the earliest known isolate, it is not necessarily the most ancestral/closest to the last common ancestor. In other words, the first detection of a virus is not always the first actual spillover of a virus, although it can be. It would be informative to compare your sequences to each representative subclades within the H5N1 isolates from cattle. E.g. A255V may have expanded in the lineage of A/bovine/Texas/24-029328-01/2024 and was not found in the last common ancestor within the clade of cattle H5N1 viruses.

• Please note that there is significant reassortments for clade 2.3.4.4b. Comparisons of mutations is only useful if the segments share the same lineage. Since the cattle outbreak appears to be a result of a single spillover and subsequent farm-to-farm spread, this analysis is appropriate. This may not be the case if reassortments are detected in the phylogenetic analysis.

• Figure 1. I noticed that there are areas in the genome where coverage drops precipitously. What are possible explanations of this observation? Failure of a primer set to amplify its target?

• Figure 2. Please indicate where the samples sequenced are in the phylogenetic tree

• For future reference, use of concatenated genome comparisons works for influenza given that no reassortment has occurred as is the case for the cattle H5N1 outbreak.

• Fig S1. Please use a different color scheme for 2.3.4.4 clades. It is difficult to distinguish a/b/c/d/e/f/g/h

• Table S3. There is one entry (dholab_carton_0028) that has a Ct value of 68.48. This looks like a typo.

**Do you want your identity to be public for this peer review?** For information about this choice, including consent withdrawal, please see our Privacy Policy

Reviewer #1: No

Reviewer #2: No

Reviewer #3: No

Reviewer #4: No

Reviewer #5: No

Reviewer #6: No

Reviewer #7: No

---

## [Author Response · Author response to Decision Letter 0]

14 Mar 2025

[Please note that this is the contents of the response to reviewers attachment without formatting. We include this here because something is required in this space, but the attached document will be much more readable.]

Dear Dr. Ayi Vandi Kwaghe,

We would like to thank the editor and reviewers for their helpful comments, which have strengthened the manuscript and allowed us to make it more clear. We are excited to send you the revised manuscript, titled "Pasteurized retail dairy enables genomic surveillance of H5N1 avian influenza virus in United States cattle." We hope that it is now acceptable for publication in PLOS ONE.

Journal Requirements:

We have reformatted the manuscript to match the requirements.

Thank you for stating the following in the Competing Interests section:

“I have read the journal's policy and the authors of this manuscript have the following competing interests: AJL has received travel funding from Oxford Nanopore Technologies to present this work at a conference.”

Please confirm that this does not alter your adherence to all PLOS ONE policies on sharing data and materials, by including the following statement: "This does not alter our adherence to PLOS ONE policies on sharing data and materials.”

AJL has received travel funding from Oxford Nanopore Technologies to present this work at a conference. This does not alter our adherence to PLOS ONE policies on sharing data and materials.

Please include captions for your Supporting Information files at the end of your manuscript, and update any in-text citations to match accordingly.

These captions have now been added.

Please review your reference list to ensure that it is complete and correct... Any changes to the reference list should be mentioned in the rebuttal letter that accompanies your revised manuscript.

We have updated several references. Preprints by Wallace et al. [1], Hu et al. [2], and Spackman et al. [3] have been published in journals, and their references have been updated to reflect this. Reviewer #6 pointed us towards the need to cite Caserta et al. [4] The same reviewer also suggested we perform a Spearman correlation, and we cited the tool we used for this [5]. A comment from reviewer #2 revealed that we had omitted citations to Rehman et al. [6] and the CDC's H5N1 Genetic Changes Inventory [7]. Those citations are now included in the manuscript. Additionally, we updated our citation of the USDA's plans for nationwide milk testing now that those policies are formalized [8].

Reviewers' Comments:

Reviewer #1:

It is not clear from the data presented what the sensitivity and specificity of the technique might be. This information is needed to assess the value of the protocol from a diagnostic perspective.

We agree that a rigorous treatment of sensitivity and specificity would be beneficial, but it must be noted that this is not a diagnostic protocol. We cannot recommend using this method for clinical samples. We perform RT-qPCR as a way to prioritize samples to sequence. If we need a diagnostic qPCR for a sample, we reach out to colleagues at the Wisconsin Veterinary Diagnostic Lab, where they have systems in place to perform diagnostic assays. Anecdotally, every sample we have sent for confirmation with the WVDL has been confirmed, so we are confident that our assay is not producing high numbers of false positives.

As far as the sensitivity of the sequencing method itself, the lowest copies/mL sample we were able to sequence well had 291 copies/mL of dairy product. We continue trying to sequence samples with lower amounts of RNA to find the lower limit of our sequencing approach.

Please clarify the statement '....Human infections have also been observed in poultry workers after presumed spillover of HPAI into poultry from cattle, underscoring the zoonotic potential of this outbreak (9).- currently it does not make sense and is not supported by the reference https://www.cdc.gov/mmwr/volumes/73/wr/mm7334a1.htm.

We have changed the wording of the sentence to "Poultry workers have also been infected, underscoring the ability of this virus to cross species barriers." (line 55)

As presented, this study does not fully align with a 'One Health approach' which is usually considered to be a transdisciplinary integrated approach to understanding a complex problem vs a focused study on a technique applied to detecting a potential pathogen which is zoonotic.

We appreciate this feedback and we agree that this study is only a start towards a true One Health approach. However, our approach is designed to monitor animal health through retail food products, which draws a clear connection to the One Health concept that human health is connected to environmental and zoonotic factors. In developing our workflow, we worked with non-academic regulators on the human and animal health fronts to understand the full scope of this complex problem.

There is mention in the conclusions about further sampling of pig farms to detect AIVs but no real consideration of how such data could be used in epidemiological studies and risk assessments. Can this be further discussed ?

We have added to the discussion (lines 445-455) to include further emphasis on these points.

Reviewer #2:

There seem to be discrepancies between numbers in the text and tables. While it is clear that the authors have tested 66 cartons in 2 months, positives are described as coming from 5 states in the abstract/text but seem to come from just 3 in Table 2. While 18 cartons are positive according to table S3, 10 are positive in Table 2 and 13 described as positive in the text. Please clarify. The Ct value indicated for carton 28 (68.48) also seems off.

Well noted. There were 66 cartons tested, and 13 of those were positive. Those 13 included samples from 5 different states. We apologize for the confusion caused by Table S3, which was taken directly from our github repository. Edits have been made to remove redundancy in instances where we tested the same carton multiple times, and to place samples that were tested in duplicate via different assays near each other. There are 13 positive cartons as tested by RT-qPCR. Three of the samples (cartons 19, 27, and 41) failed pre-sequencing quality control and sequenced poorly during our method development. We now specifically name those samples in the text (line 265). Unfortunately, these three samples accounted for two of the states with positives. Table 2 includes the 10 samples (from 3 states) that we attempted to sequence in this study.

The erroneous Ct value should not have been included in the table. This sample was inconclusive, with one positive replicate and the other negative. We have decided to not report inconclusive samples, so it has been removed.

How did you set your thresholds? Both for the 20x sequence coverage chosen and for the 10% frequency for the variants search.

For the 20% coverage threshold, we wanted to make sure that we chose a threshold that was well above the coverage we saw in the negative controls. While a large majority of the negative controls had less than 10x coverage, there were a few regions that exceeded 10x. None of the coverage ever exceeded 20x in our negative controls, which gives us confidence that any coverage above that threshold is real and meaningful data from our samples.

For the variant frequency threshold, we wanted to have a frequency that reflected multiple reads. We did not want to report variants as being present if there was only one read that supported that variant. Assuming the coverage is the lowest it can be (20x) two reads would be 10% frequency. While it is likely that many positions are covered more deeply than the minimum, we still felt like a 10% cutoff allows us to analyze low-frequency variants without including spurious signals.

Mammalian adaptation markers are known. It is a pity that the authors did not screen their sequence data for those (irrespective of the variants frequency actually, it would be quite informative in a One Health perspective as stressed in the Introduction and Discussion sections).

Thank you for pointing out this omission. We did compare our observed mutations to a list of known adaptation and virulence markers in influenza, but we should have reported the lists of virulence markers we used. We used tables presented in Nguyen et al., Rehman et al., and the CDC's H5N1 Genetic Changes Inventory. These citations are now added to the references for this manuscript. We reported the results of the screening in the results section under "Few major amino acid substitutions were detected" (lines 336-340). We chose to avoid replicating those previous publication's lists when we only detected a handful of variants.

Figure 2: the tree is not very informative. What do divergence percentages refer to? What is the scale? The Figures legends are very short and would deserve more info.

We have expanded on the figure legends to address these concerns. In Figure 1, we added an explanation about the coverage drop-off at certain amplicons. For Figure 2, we have explained the divergence metric and mentioned that gray points have missing metadata.

Reviewer #3:

Authors may improve the discussion section by discussing the global implications of the findings and how the approach could be adapted for use in other regions or for monitoring other zoonotic pathogens.

We have added this sentence in the discussion: "In fact, sampling retail milk to detect cattle outbreaks could be implemented in any area where the production region of milk is available at the point of sale, not just in Canada and the US." (lines 442-444)

We also added more detail into our discussion section addressing swine flu monitoring (lines 445-455), but the pros and cons of our methods in that discussion are broadly applicable to any novel implementation of this approach.

Also, they have to clearly articulate the limitations of using retail dairy for surveillance, such as sample representativeness, phylogenetic challenges, and potential for false negatives.

We addressed sample representativeness on line 383, writing "This reduces the effectiveness of retail milk sampling for surveillance purposes because some farms and dairy processors produce little or no product that could be sampled as retail fluid milk." Addressing phylogenetic challenges, we write on line 396: "This limits the utility of phylogenetic analyses, which assume that each sequence represents a single infected individual." With respect to the potential for false negatives, we have added a few sentences on lines 391-394 explaining this further - "While a sample that tests positive from an area is an indicator of H5N1 infection, the reverse is not true. There are many reasons why H5N1 infections may not be picked up by this sampling strategy, so a negative test from a given area is not a declaration that influenza is absent in that area." We hope these edits address these concerns.

Reviewer #4:

Reviewer #4 left no points to address.

Reviewer #5:

1. Paragraph 6, you don’t need to mention “the home state of the researchers on this study”, instead you might need to give more other interesting point about Wisconsin besides “the second- largest dairy producing state in the US”.

We have removed the reference to Wisconsin as the home state of the researchers.

2. It is better to put a reference about the existing workflow from your work in SARS-CoV-2 sequencing, as you mentioned it in introduction.

We now refer to QIAseq Direct on line 88 in the introduction.

3. It is better to give reference to “OHHLEP’s 4 Cs” strategy which you also mentioned it in introduction.

We have added a reference to OHHLEP's 4 Cs in the introduction on line 91.

4. You mentioned in “sample collections” about the dairy products which you took as sample, can you explain about the percentage of each sample taken? How many percent of whole milk did you take and how many percent low-fat wilk did you take? Etc.

We aimed to sample whole milk (4% cream) if it was available. Approximately 50% of our samples were whole milk, while increasingly small numbers were cream (7 samples), skim milk (7 samples), lowfat milk (5 samples), and half and half (4 samples). We got either one or two samples of yogurt, chocolate milk, kefir, and buttermilk.

5. It is unclear about how you finally only mentioned about samples from 10 sample cartons from 66 samples which you previously mentioned.

See previous response to reviewer #2's question about the same subject. We hope this has become clearer to the reader in the revised manuscript.

6. Why did the observation only do in cartons 21,24, 46,63 and 65? Because we could see that segments’ coverage on cartons 5 and 6 are also over than 90%.

Thank you for bringing up this point. We have made changes in the paragraph (lines 257-267) that describes passing segments to clear up the wording and correct errors.

7. The discussion part is more like simple introduction rather than a discussion of result which you have.

We covered the limitations of this study as well as its future directions in the discussion, but we have also now added some to the discussion surrounding the mutations we were able to identify (lines 372-374).

Reviewer #6

The abstract could use a more comprehensive summary of results – e.g. qPCR results, more description of genome coverage (significant correlation with qPCR result?) and/or phylogenetic clustering of resulting sequences.

We have added to the abstract: "...with higher viral copies corresponding to better sequencing success. The sequences clustered phylogenetically within the rest of the cattle outbreak sequences reported."

A Spearman correlation should be performed to support the following claim “The method yielded an overall average sequencing depth of 8,176x across all samples, with individual samples varying from 325x to 29,256x. This correlated with the input vRNA amount, with less input vRNA yielding lower depth in sequencing.”

Thank you for this suggestion. We performed a Spearman's correlation on the aforementioned claim and found that the claim is well-supported. The correlation coefficient was 0.946, with a p-value of 3 x 10-7. We have added this result to the manuscript on lines 285-287.

The quality of the Fig. 2 phylogeny needs to be improved for readability/interpretation. If a tree at higher resolution with less node crowding cannot be made in Auspice, recommend downloading the tree and metadata files from Auspice and re-creating the tree. Several tree tips in Fig. 2 are white/gray – do these not have location metadata? Besides trait inference, did sequenced samples cluster with other specimens from the state? It appears possible for Michigan samples but not Colorado but given the low quality/crowding in the tree is difficult to see. A description of clades that sequenced samples fall into may help.

We have remade Figure 2 based on your suggestion to reduce vertical crowding of the points. We attempted to generate a new tree in TreeViewer and iTol, but we found that the number of points was not conducive to display in either of these programs. With 1626 points, there must be some node overlap to allow the tree to fit on a single, readable page. With this in mind, we have changed the aspect ratio of Figure 2 to allow the points more room to spread out vertically, which we hope improves the readability of the tree. As far as quality, the figure that was uploaded to PLOS ONE (and that we can download from our submission portal) was of much higher quality than the one included in the pdf manuscript. The loss in quality seems beyond our direct control but we hope the editors can propagate a high resolution version when circulating this revised version for review.

To address the second part of your question, our samples from Michigan fell into a clade composed of 188 sequences from Michigan, Ohio, and Texas. Our samples from Colorado fell into the clade defined by NA:N71S, which includes 1163 sequences from across the country. Cartons 46 and 65 clustered more closely than any others, right next to 3 sequences from the Broad institute's retail milk sampling of Colorado. These sequences are within

---

## [Decision Letter · Decision Letter 1]

29 Mar 2025

Dear Dr. Lail,

Thank you for submitting your manuscript to PLOS ONE. After careful consideration, we feel that it has merit but does not fully meet PLOS ONE’s publication criteria as it currently stands. Therefore, we invite you to submit a revised version of the manuscript that addresses the points raised during the review process.

We look forward to receiving your revised manuscript.

Kind regards,

Ayi Vandi Kwaghe, D.V.M., M.V.Sc., P.G.D.E. Ph.D., MPH, FETP

Academic Editor

PLOS ONE

Journal Requirements:

Reviewers' comments:

Reviewer's Responses to Questions

**Comments to the Author**

Reviewer #1: All comments have been addressed

Reviewer #5: All comments have been addressed

2. Is the manuscript technically sound, and do the data support the conclusions?

Reviewer #1: Yes

Reviewer #5: Yes

3. Has the statistical analysis been performed appropriately and rigorously?

Reviewer #1: N/A

Reviewer #5: Yes

4. Have the authors made all data underlying the findings in their manuscript fully available?

Reviewer #1: Yes

Reviewer #5: (No Response)

5. Is the manuscript presented in an intelligible fashion and written in standard English?

Reviewer #1: Yes

Reviewer #5: Yes

Reviewer #1: This added section could be more clearly written i.e. ……..Swine are often considered a mixing vessel for recombination between bird, human, and swine flu strains (add reference). This is reflected in the strict biosecurity protocols in use on pig farms. Just like dairy farms, these operations have retail outputs (such as meat ?) which may prove helpful to monitoring these viruses. Performing (asymptomatic?) genomic surveillance (on healthy/asymptomatic animals) at the population level by sampling outputs (meat?) from these locations may (generate data to) help inform risk assessments. However, it is unlikely that this approach (i.e testing milk samples to monitor H5N1 in dairy cattle ?) can be simply reconfigured to swine or another system given the need for quick turnaround and traceable supply chains (please clarify what is meant here ). Dairy ? (Milk ) is pooled between many animals, often sold within a month of production, and contains a marker of where it was produced. It's possible that extending monitoring to other agricultural settings would require orthogonal (please clarify what is meant here) approaches like wastewater or air monitoring..’

Reviewer #5: (No Response)

**Do you want your identity to be public for this peer review?** For information about this choice, including consent withdrawal, please see our Privacy Policy

Reviewer #1: No

Reviewer #5: No

---

## [Author Response · Author response to Decision Letter 1]

28 Apr 2025

Based on Reviewer 1's comments, we have edited this section to be more clearly written: “Swine are often considered a mixing vessel for recombination between bird, human, and swine flu strains [51]. This is reflected in the strict biosecurity protocols in use on pig farms. Just like dairy farms, these operations have retail outputs, like meat, which may prove helpful to monitoring these viruses. Performing viral genomic surveillance on asymptomatic animals at the population level by sampling outputs from these locations may generate data to help inform risk assessments. However, it is unlikely that the same approach used for milk samples in this study can simply be reconfigured to swine or another system. There is a need for quick turnaround and traceable supply chains to produce geographically informed, actionable information. Milk is pooled between many animals, often sold within a month of production, and contains a marker of where it was produced. It is possible that extending population level monitoring to other agricultural settings would require alternative approaches like wastewater or air monitoring.”

---

## [Editor Report · Decision Letter 2]

9 May 2025

Amplicon sequencing of pasteurized retail dairy enables genomic surveillance of H5N1 avian influenza virus in United States cattle

PONE-D-24-59493R2

Dear Dr. O'Connor,

We’re pleased to inform you that your manuscript has been judged scientifically suitable for publication and will be formally accepted for publication once it meets all outstanding technical requirements.

Kind regards,

Ayi Vandi Kwaghe, D.V.M., M.V.Sc., P.G.D.E. Ph.D., MPH, FETP

Academic Editor

PLOS ONE
---

## [Editor Report · Acceptance letter]

PONE-D-24-59493R2

PLOS ONE

Dear Dr. O'Connor,

I'm pleased to inform you that your manuscript has been deemed suitable for publication in PLOS ONE. Congratulations! Your manuscript is now being handed over to our production team.

Kind regards,

on behalf of

Dr. Ayi Vandi Kwaghe

Academic Editor

PLOS ONE